# Nearly one-third of lactating mothers are suffering from undernutrition in pastoral community, Afar region, Ethiopia: Community-based cross-sectional study

**Getahun Fentaw Mulaw**[1]*, **Kusse Urmale Mare**[2], **Etsay Woldu Anbesu**[3]

**1** School of Public Health, College of Health Sciences, Woldia University, Woldia, Amhara Region, Ethiopia,
**2** Nursing Department, College of Health Sciences, Samara University, Samara, Afar Region, Ethiopia,
**3** Public Health Department, College of Health Sciences, Samara University, Samara, Afar Region, Ethiopia

* gechfentaw1014@gmail.com

## Abstract

### Background

Undernutrition is responsible for a significant proportion of maternal and child morbidity and mortality. Lactating women are nutritionally vulnerable groups because this period places a high nutritional demand on the mother and leads to nutritional stress. Poor nutrition during lactation has a significant negative consequence to mothers and children's survival, growth, and development. Therefore this study aimed to assess the nutritional status of lactating mothers and associated factors in pastoral community, Afar region, Ethiopia.

### Methods

A community-based cross-sectional study was conducted from January 5/2020 to February 10/2020, in the Abala district. The data were collected from a sample of 366 lactating mothers whose children aged less than 24 months. Data was collected through face-to-face interviews and anthropometric measurements. Study participants were recruited using a systematic sampling technique. Anthropometric measurements (both body mass index and mid-upper arm circumference) were taken from each mother using calibrated equipment and standardized techniques. Data were entered into Epi-data version 4.2 and exported to SPSS version 22 for analysis. Predictor variables with a P-value < 0.25 at bivariable analysis were candidates for the final model. Statistical significance was declared at P-value of < 0.05 in the multivariable logistic regression.

### Result

This study showed that 120(32.8%) and 122(33.3%) surveyed mothers were undernourished using a cut-off body mass index <18.5 kg/m$^2$ and mid-upper arm circumference <23 Centimeter, respectively. Lactating mothers who didn't meet the minimum dietary diversity score were more than five (Adjusted odds ratio (AOR) = 5.103; 95% confidence interval (CI): 2.128, 12.238) times more likely to be undernourished than those who met the

**Data Availability Statement:** All relevant data are within the manuscript and its Supporting Information files.

**Funding:** The funder (Samara University) has no role in study design, data collection and analysis, decision to publish, or preparation of the manuscript.

**Competing interests:** The authors have declared that no competing interests exist.

minimum dietary diversity score. Mothers with short birth intervals were also more than four (AOR = 4.800; 95% CI: 2.408, 9.567) time more likely to be undernourished.

## Conclusion

Nearly one-third of lactating mothers were undernourished. Undernutrition among lactating mothers was significantly associated with maternal dietary diversity score and birth interval. Health education on proper and adequate maternal dietary feeding practices, and proper family planning utilization during lactation should be emphasized.

## Background

Malnutrition is one of the most devastating problems worldwide and can be manifested as both undernutrition and overnutrition. Undernutrition can be divided into protein-energy malnutrition and micronutrient deficiencies [1]. Undernutrition is the outcome of insufficient food intake, inadequate care, and infectious diseases [2, 3]. Malnutrition among reproductive-age women have a major impact on their own and their children's health [4, 5].

Lactating women are nutritionally vulnerable groups because this period places a high nutritional demand on the mother and leads to nutritional stress [6–8]. The diet consumed by the lactating mother will help to fulfill their own nutritional needs and prevents depletion of their body stores, and also enable them to produce enough milk for their infants and children [9–11]. Poor nutrition during lactation poses a significant threat to mothers and children's survival, growth, and development [12–14].

In lower and lower-middle income countries, undernutrition is responsible for a significant proportion of maternal and child morbidity and mortality [10]. Globally, around 9.4% of adult women were affected by undernutrition [15]. In African countries, the figure of underweight in reproductive-age women ranges from 5–20% [16, 17]. In Ethiopia, the national prevalence of undernutrition among reproductive-age women, as measured by BMI $<18.5 kg/m^2$, is 22.4%, while in the Afar region it is 39.1% [18]. Other mini-review done on lactating mothers also identified that; the prevalence of undernutrition is highest (50.6%) in the northern parts and lowest (17.4%) in the southern parts of Ethiopia [19].

Despite the number of effective and proven nutrition intervention programs in Ethiopia [20, 21], studies reported that a significant proportion of lactating mothers are undernourished [22–24]. Evidence-based, timely, and contextualized information on the nutritional status and related factors is vital to review those programs and strategies.

Even though there are a limited number of studies conducted to assess the nutritional status of lactating women in different regions of Ethiopia, there is no documented study done in the Afar region. Therefore, this study aimed to assess the nutritional status of lactating mothers and associated factors in the pastoral community, Afar region, northeast Ethiopia.

## Methods and materials

### Study design, setting, and period

A community-based cross-sectional study was conducted in the Abala district. It is one of the districts in zone two of the Afar region, located at the base of the eastern escarpment of the Ethiopian highlands about 942 kilometers northeast of Addis Ababa and 491 km far from the regional capital city, Samara. According to the projection of the 2007 national Census, the district has a total population of 43,372 with an area of 1,188.72 square kilometers. The district

had one general hospital, four health centers, and eight functional health posts [25]. The study period was from January 5/2020 to February 10/2020.

## Eligibility criteria

The source population was all lactating women whose children aged less than 24 months in the Abala district. Mothers were included if they lived in the district for six or more months. Mothers were considered to be excluded if they were seriously ill and physically unable to fit for anthropometric measurements, but no lactating mother was found with such cases. Additionally, women who were or who suspect pregnancy, according to maternal reports, were excluded from the study.

## Sample size determination

**The sample size for the first objective (for assessing the prevalence of undernutrition).**   The sample size was calculated using a single population proportion formula with a 95% confidence level, 5% margin of error, 25% estimated prevalence of undernutrition (chronic energy deficiency) in the study done in Tigray region [24], and by considering a non-response allowance of 10%. The calculated sample size for assessing the first objective was 317.

**The sample size for the second objective (to determine factors associated with undernutrition).**   The sample size was determined using Epi info version 7 considering the assumptions of 95% CI, 80% power, the ratio of unexposed to exposed 1:1, and assuming different predictors from researches conducted in Ethiopia [26–28]. The predictor variable nutritional advice gives the largest sample size 370 (Table 1).

Therefore, the final sample size for this study, to determine both objectives, was taken as 370. This is a minimum representative sample for the study population.

## Sampling procedure

The sampling procedure was a multi-stage sampling. The Abala district has 14 kebeles from which 5 kebeles were selected randomly. In each selected kebeles, the list of all households having lactating mothers was located (from the health extension worker's family folder). Then the sample size was proportionally allocated based on the total number of lactating women in each selected kebele. Finally, study participants in each selected kebeles were recruited from the list using a systematic sampling technique, every fixed interval. In households where there are more than one lactating women, a lottery method was used to select one participant (Fig 1).

## Data collection procedure and quality control measures

A semi-structured questionnaire was developed through a critical review of relevant literature [6, 19, 22–24, 26, 29–34]. The questionnaire had the following parts, namely socio-demographic characteristics, maternal health care practice, maternal feeding practice, sanitation and

**Table 1.  Predictor variables selected for determining sample size for assessing factors associated with nutritional status of lactating mothers in Abala district, Afar, Ethiopia.**

| No | Significant predictors | Citation | CI | Power | Cases: control | Percent of exposure (%) | | Samples size including 10% non-response rate | | |
|---|---|---|---|---|---|---|---|---|---|---|
| | | | | | | Cases | Controls | Cases | Controls | Total |
| 1. | Marital status | [26] | 95 | 80 | 1:1 | 62.00 | 14.93 | 20 | 20 | 40 |
| 2. | Nutrition advice | [27] | 95 | 80 | 1:1 | 14.90 | 27.30 | 185 | 185 | **370** |
| 3. | Meal frequency | [28] | 95 | 80 | 1:1 | 61.76 | 35.13 | 62 | 62 | 124 |

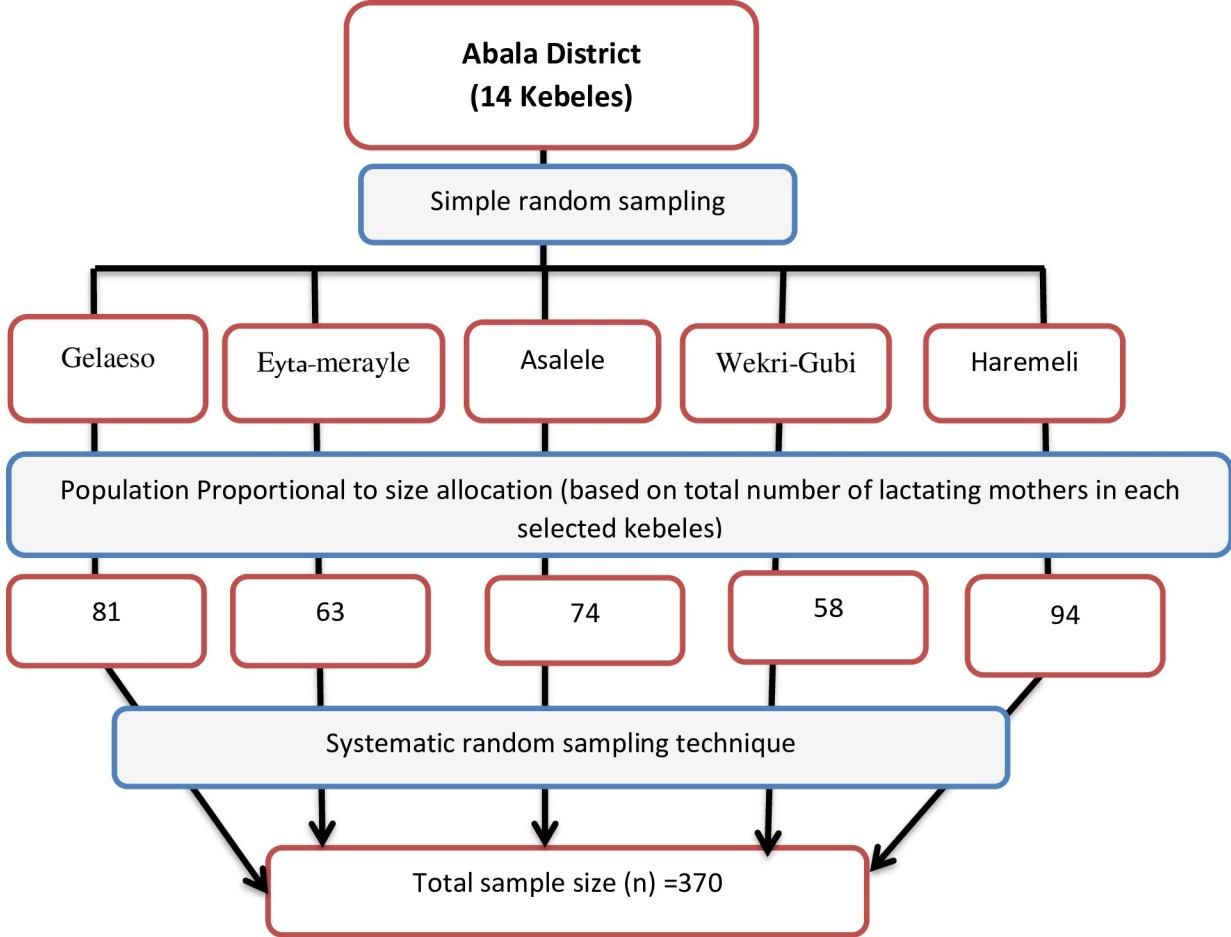

**Fig 1. Schematic representation of sampling methods and procedures to select study participants, in Abala district, Afar region, northeast Ethiopia, 2020.**

hygiene-related factors, and anthropometric measurements. The research questionnaire was prepared in the English and translated into the local language. The questionnaire was pre-tested and validated on 10% of the sample size in the none-selected kebeles of the Abala district. Data was collected by direct face-to-face interviewing with lactating mothers and measuring anthropometry.

Six diploma-holder health professionals as data collectors and two master-holder public health professionals as supervisors were recruited. They were selected based on their fluency in the local language and they were trained on data collection techniques.

The dietary diversity of the lactating women was assessed using the food and agricultural organization of United States (FAO) women's dietary diversity score (WDDS) assessment scale. It was calculated by a simple count and summing-up of the number of food groups an individual respondent consumed over the preceding 24-hours recall period regardless of the portion size. The WDDS were calculated using the nine food groups (starchy staples, legumes and nuts, milk and milk products, dark green leafy vegetables, other vitamin-A rich fruits and vegetables, other fruits and vegetables, organ meat, meat and fish, and eggs). The study participants were asked whether they ate or not from each food group 24-hours preceding the survey. The cut-off point for the minimum dietary diversity score (MDDS) is the consumption of at least four or more food groups [35].

The anthropometric measurements such as weight, height, and mid-upper arm circumference (MUAC) of the lactating mother were performed according to the world health organization (WHO) standardized procedures. The weight of lactating mothers was measured using the United Nations International Children's Emergency Fund (UNICEF) Electronic Scale with a maximum capacity of 150 kg and an accuracy of 0.1 kg. Scales were calibrated and standardized every morning before the start of measuring the mother's weight, using a 5 kg standard weight. Mothers' weight was measured with light clothing and without shoes.

Maternal height was measured with standing position using a portable stadiometer to the nearest 0.1 cm, by allowing the mothers to take off their shoes and standing up straight placing their back against the wall. Maternal MUAC was measured using UNICEF measuring tape to the nearest 0.1 cm by placing WHO MUAC measuring tape loosely on the upper-middle arm between the acromion and olecranon process of the non-dominant hand.

Continuous supervision and follow-up of the data collectors were made to review and check for completeness and consistency of the collected data on daily basis. The collected data was handled and stored carefully and appropriately.

## Data processing and analysis

The raw data was coded, cleaned, edited, and entered into Epi-data version 4.2 and transferred to statistical package for social sciences (SPSS) version 22 for analysis. The result of categorical variables was presented using frequency and percentage, bar graph and pie charts, and that of continuous variables with a mean (SD). For continuous variables, normality was checked using graph (histogram) and statistics (Kolmogorov–Smirnov Test).

Bivariable and multivariable logistic regressions were used to assess the association of predictor variables with the outcome variable. Significance was determined using the adjusted and unadjusted odds ratio with 95% CI and P-value. All predictor variables that have an association with the outcome variable at bivariable analysis with the P value of $\leq 0.25$ were selected and included in the multivariable logistic regression model. Variables with a P-value $< 0.05$ in multivariable analysis were declared as statistically and independently significant predictors of undernutrition among lactating mothers.

The final model was tested using Hosmer and Lemeshow's chi-square test p-value, and the P–value was 0.556, which showed that the model was the best fit. The percentage of the model that was accurately classified was 74% and the extent of multicollinearity was also assessed using standard error cut-off two, but there was no multicollinearity among studied variables.

## Ethical consideration

The study protocol was approved by the Institutional Ethical Review Committee of Samara University; College of Health and Medical Science (ERC0053/2019). Written informed consent was obtained from lactating mothers, and confidentiality was also assured.

## Operational definitions/definition of terms

- **Maternal nutritional status**:—is measured using either BMI or MUAC. The mother was considered as

  ○. Undernourished: if her BMI $< 18.5$ Kg/m$^2$ and/or MUAC$< 23$cm

  ○. Not undernourished: if her BMI $\geq 18.5$ kg/m$^2$ and/or MUAC $\geq 23$ cm

- **Women dietary diversity score (WDDS):- is calculated using the** nine food groups' category [35, 36]. The mother was asked the type of food she ate 24-hours preceding the survey regardless of the portion size. WDDS is considered as

  ○ **Low**: if the mother consumed ≤ 3 food groups out of the nine food groups

  ○ **Moderate**: if the mother consumed 4–5 food groups out of the nine food groups

  ○ **High**: if the mother consumed ≥ 6 food groups out of the nine food groups

- **Minimum Meal frequency (MMF):-** is the frequency or number of meals served by lactating mother 24-hours preceding the survey. The recommended energy intake during the first 6 months of lactation is an additional 500 kilocalorie, which can be achieved by a dense extra meal per day [37, 38].

  ○ **Meet:** ≥ 4 times

  ○ **Not meet:** < 4 times

- **Lactating mother**: A mother who was breastfeeding her child at the time of the survey

## Result

### Socio-demographic characteristics of study participants

A total of 366 lactating mothers were included in this study, with a response rate of 98.9%. The mean (±SD) ages of study participants were 29.6(±7) years. The majority, 298(81.4%) and 311 (85%) of them were rural residents and Afar in ethnicity, respectively. Three-fifth, 223(60.9%) of mothers were not attending formal education. The mean (±SD) of the study participant's family size was 5.6(+2.2), and the mean age of the index children was 11.5(±5.9) months (Table 2).

### Maternal healthcare-related characteristics

Less than two-third, 229(62.6%) of mothers were attended antenatal care (ANC) follow-up. One hundred five (28.7%) of the study participants conceive their first pregnancy during their teenage. More than half, 198(54.1) of participants had less than 36 months of preceding birth interval. Nearly three-fourth, 265(72.4%) of the participants gave birth, to the index child, at home (Table 3).

### Maternal nutritional status

Maternal nutritional status was assessed using both BMI and MUAC. According to BMI, 120(32.8%) of mothers were undernourished; with mild, moderate, and severe degrees of undernourishment occurred in 80(21.9%), 30(8.2%), and 10(2.7%) of respondents, respectively. But using MUAC, 122 (33.3%) of mothers were undernourished. The Pearson's correlation coefficient(r) between BMI and MUAC was 0.764 (P-value <0.001). The mean (±SD) score of mothers BMI, and MUAC were 20.2(±2.8) kg/m2, 23.9(±2.5) cm, respectively (Table 4 and Fig 2).

### Maternal feeding practice

Only 34(9.3%), and 114(31.1%) of mothers received at least one extra meal intake during pregnancy and lactation per day, respectively. Less than one-fourth, 88(24%) of mothers met the MDDS, while 120(32.8%) of them met the minimum meal frequency. Based on the 24-hours

**Table 2. Socio-demographic characteristics of study participants (n = 366) in the Abala district, Afar region, northeast Ethiopia, 2020.**

| Variables | Category | Frequency (n) | Percentage (%) |
|---|---|---|---|
| Maternal age | 15–24 | 107 | 29.2 |
| | 25–34 | 161 | 44.0 |
| | 35–49 | 98 | 26.8 |
| | Mean (±SD) | 29.6 ± 7.0 | |
| Residence | Rural | 298 | 81.4 |
| Ethnicity | Afar | 311 | 85.0 |
| | Others* | 55 | 15.0 |
| Religion | Muslim | 312 | 85.2 |
| | Orthodox | 54 | 14.8 |
| Maternal marital status | Married | 360 | 98.4 |
| | Unmarried** | 6 | 1.6 |
| Maternal educational status | Not attended formal education | 223 | 60.9 |
| | Attended 1° education | 106 | 29.0 |
| | Attended 2 and above education | 37 | 10.1 |
| Maternal occupation | Pastoralist | 168 | 45.9 |
| | Housewife | 120 | 32.8 |
| | Merchant | 50 | 13.7 |
| | Employed*** | 28 | 7.6 |
| Paternal educational status (n = 360) | Not attended formal education | 228 | 63.3 |
| | Attend 1° education | 60 | 16.7 |
| | Attend 2° and above education | 72 | 20.0 |
| Paternal occupation (n = 360) | Pastoralist | 235 | 65.3 |
| | Employed*** | 69 | 19.2 |
| | Others **** | 56 | 15.5 |
| Family size | ≤ 4 | 143 | 39.1 |
| | 5–8 | 181 | 49.4 |
| | ≥ 9 | 42 | 11.5 |
| | Mean (±SD) | 5.6(±2.2) | |
| Number of U-5 children within the household | One child | 113 | 30.9 |
| | Two or more children | 253 | 69.1 |
| Time taken to reach to Near-by Health Institution | <15 minute | 173 | 47.3 |
| | 15–30 minute | 188 | 51.3 |
| | >30 minute | 5 | 1.4 |
| Decision maker of the household | Mostly father | 252 | 68.9 |
| | Mostly mother | 63 | 17.2 |
| | Jointly | 51 | 13.9 |
| Owing Television and/or Radio (yes) | | 60 | 16.4 |
| Farming land ownership (yes) | | 106 | 29.0 |
| Livestock ownership (yes) | | 299 | 81.7 |
| Sex of the child | Male | 176 | 48.1 |
| Child age (in months) | 0–5 | 73 | 19.9 |
| | 6–11 | 106 | 29.0 |
| | 12–23 | 187 | 51.1 |
| | Mean (+SD) | 11.5(±5.9) | |

*Tigray, Amhara.

**Single, divorced, widowed.

***Employed in government and non-governmental organizations.

****Merchant, daily laborer, farming.

**Table 3. Maternal healthcare-related characteristics of study participants in the Abala district, Afar region, northeast Ethiopia, 2020.**

| Variables | Category | Frequency (n = 366) | Percentage (%) |
|---|---|---|---|
| Antenatal care (ANC) | Yes | 229 | 62.6 |
| ANC frequency (n = 229) | < 4 times | 166 | 72.5 |
| | ≥ 4 times | 63 | 27.5 |
| Maternal age during first pregnancy | < 20 Years | 105 | 28.7 |
| | ≥ 20 years | 261 | 71.3 |
| Gravidity | Primi-gravida | 67 | 18.3 |
| | 2–4 | 169 | 46.2 |
| | ≥ 5 | 130 | 35.5 |
| Birth interval | First child | 67 | 18.3 |
| | < 36 months | 198 | 54.1 |
| | ≥ 36 months | 101 | 27.6 |
| Birth place | Home | 265 | 72.4 |
| | Institutional | 101 | 27.6 |
| Getting diseased in the past two months | Yes | 81 | 22.1 |
| Getting nutrition counseling | Yes | 228 | 62.3 |
| Maternal workload | Yes | 78 | 21.3 |

recall method, the mean (±SD) score of Maternal dietary diversity score and maternal meal frequency was 3.09(±0.875) and 3.33(±0.570), respectively.

Two-hundred seventy-eights (76%) of study participants had a low dietary diversity score (Fig 3). In the last 24-hours preceding the survey, all study participants consumed starchy staples, but only 16.9%, 3.8%, and 2.7% of them received dark green leafy vegetables, eggs, and organ meat, respectively (Fig 4).

## Sanitation and hygiene-related characteristics

Three hundred thirty-seven (92.1%) of the study participants got water from protected sources. Less than half, 163(44.5%) of households had a latrine, from which 155(42.3) is functional at the time of the survey. Regarding type of latrine, majority 138(84.7%) is pit type, and the rest 25(15.3) is ventilated improved pit latrine (VIPL) type. Nearly three-fourth, 271(74%) of the study participants dispose of solid waste in open-field outside their compound, while 73 (20%) and 22(6%) of the dispose in private pit, and open field in their compound.

**Table 4. Nutritional status of study participants in the Abala district, Afar region, northeast Ethiopia, 2020.**

| Variables | Category | Frequency | Percentage (%) |
|---|---|---|---|
| Maternal BMI (kg/m$^2$) | <18.5 | 120 | 32.8 |
| | 18.5–24.9 | 230 | 62.8 |
| | 25–29.9 | 15 | 4.1 |
| | ≥ 30 | 1 | 0.3 |
| | Mean (+SD) | 20.2(±2.8) | |
| Degree of undernutrition (n = 120) | Mild (BMI:17–18.49 kg/m$^2$) | 80 | 21.9 |
| | Moderate (BMI:16–16.69kg/m$^2$) | 30 | 8.2 |
| | Severe (BMI:<16kg/m$^2$) | 10 | 2.7 |
| Maternal MUAC (cm) | < 23 | 122 | 33.3 |
| | ≥23 | 244 | 66.7 |
| | Mean (±SD) | 23.9(±2.5) | |

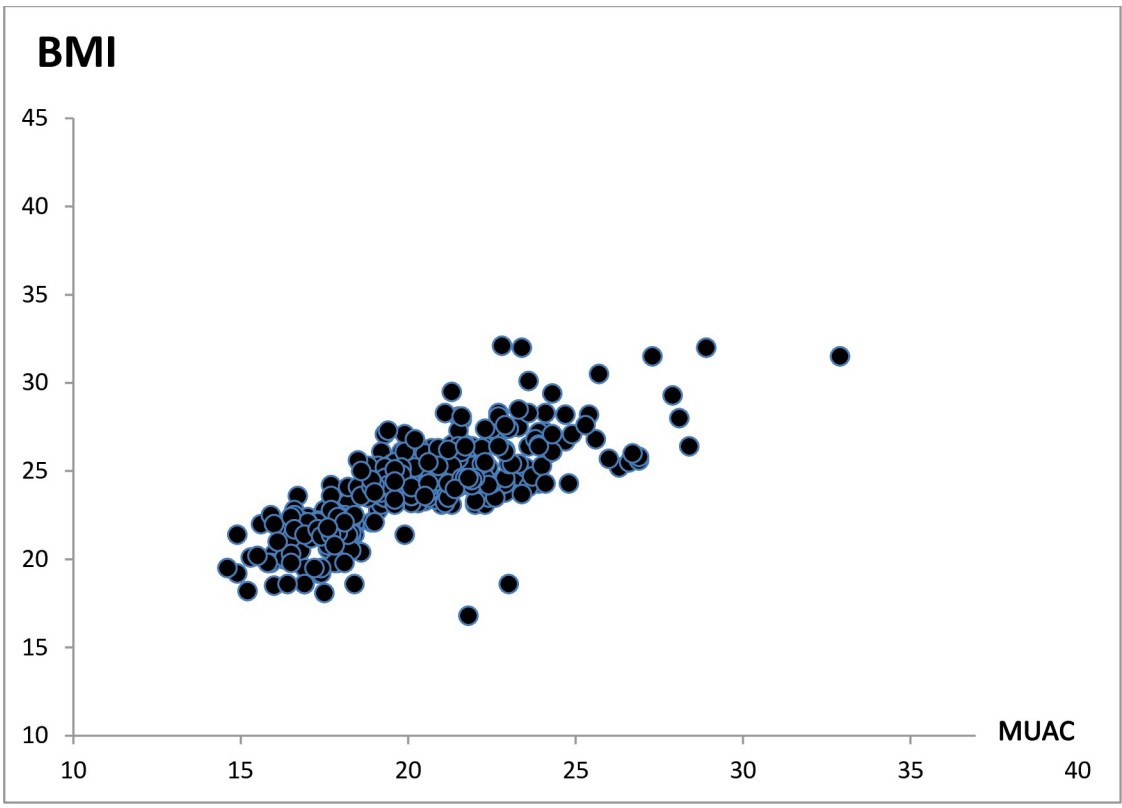

**Fig 2. Scatter plot showing the relation between maternal BMI and maternal MUAC (r = 0.764), of study participants in Abala district, Afar region, northeast Ethiopia, 2020.**

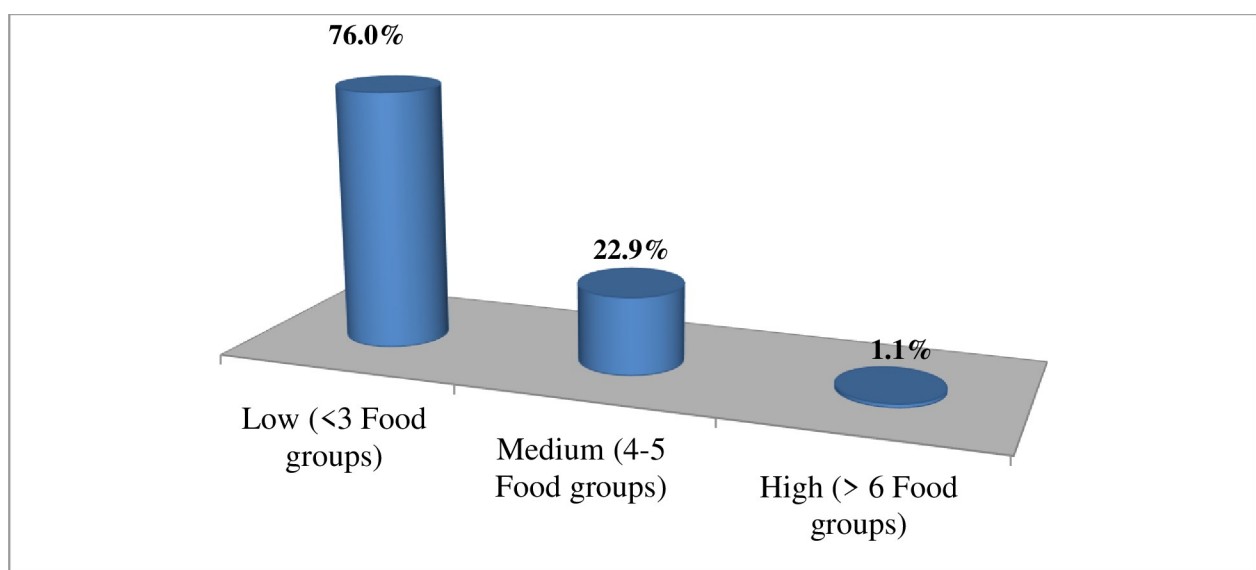

**Fig 3. Level of dietary diversity score of study participants (n = 366) in Abala district, Afar region, Northeast Ethiopia, 2020.**

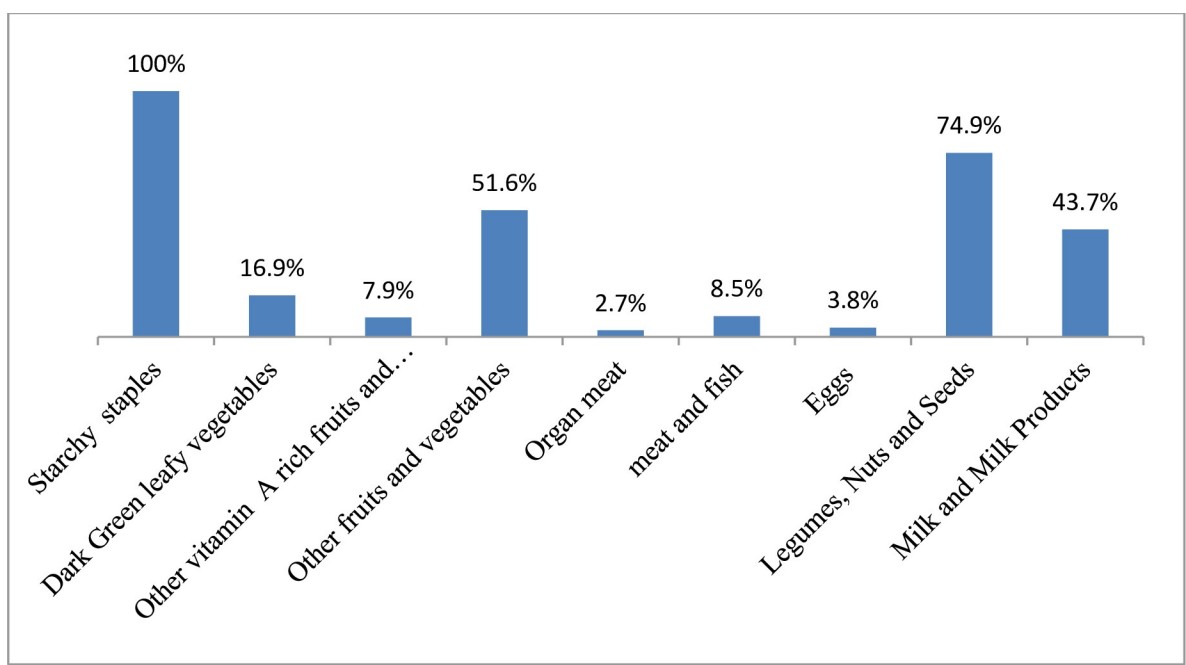

**Fig 4. Pattern of food group consumption of study participants (n = 366) in Abala district, Afar region, Northeast Ethiopia, 2020.**

All study participants, 366(100%), had the practice of hand-washing before preparing food and before feeding themselves. However, only 71(19.4%) of them used soap for hand-washing (Fig 5).

## Factors associated with undernutrition

At binary logistic regression material age and education, paternal education and occupation, farming land ownership, maternal ANC follow-up, gravidity, birth interval, getting nutritional counseling, maternal dietary diversity score, and meal frequency were associated with maternal undernutrition at p-value<0.25. But in the final multi-variable logistic regression model, maternal dietary diversity and birth interval were found statistically significantly associated (at the p-value < 0.05) with underweight among lactating mothers. Lactating mothers who didn't

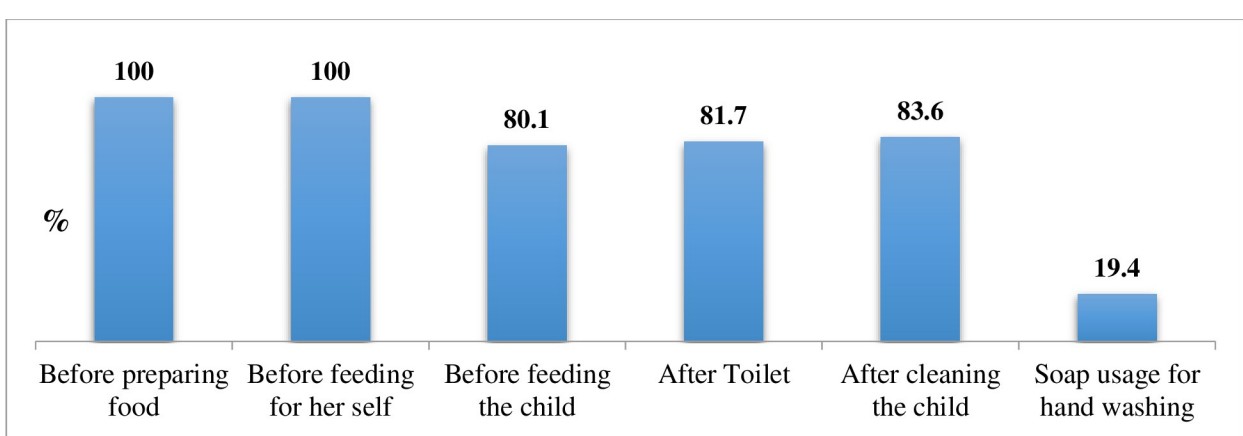

**Fig 5. Proportion of hand-washing practice of study participants (n = 366) in Abala district, Afar region, Northeast Ethiopia, 2020.**

meet the MDDS were more than five (AOR = 5.103; 95% CI: 2.128, 12.238) times more likely to be underweight than those who met the MDDS. Mothers with short birth intervals were more than four (AOR = 4.800; 95% CI: 2.408, 9.567) time more likely to be undernourished (Table 5).

**Table 5. Factors associated, at binary and multivariable logistic regression, with undernutrition among lactating mothers in the Abala district, Afar region, northeast Ethiopia.**

| Variables / Category | Undernutrition (BMI<18.5kg/m$^2$) | | COR(95% CI) | AOR(95% CI) |
|---|---|---|---|---|
| | Yes | No | | |
| **Maternal age (Years)** | | | | |
| 15–24 | 28 | 79 | 0.54(0.30, 0.97) | 0.58(0.18, 1.88) |
| 25–34 | 53 | 108 | 0.74(0.44, 1.25) | 0.85(0.37, 1.95) |
| 35–49 | 39 | 59 | 1 | 1 |
| **Maternal education** | | | | |
| Not educated | 80 | 143 | 2.03(0.89, 4.65) | 0.59(0.16, 2.15) |
| Attend primary school | 32 | 74 | 1.57(0.65, 3.80) | 0.56(0.17, 1.90) |
| Attend secondary & above | 8 | 29 | 1 | 1 |
| **Paternal education** | | | | |
| Not educated | 82 | 146 | 2.33(1.22, 4.43) | 1.96(0.57, 6.76) |
| Attend primary school | 21 | 39 | 2.23(1.01, 4.91) | 1.69(0.58, 4.92) |
| Attend secondary & above | 14 | 58 | 1 | 1 |
| **Paternal occupation** | | | | |
| Employed | 15 | 54 | 1 | 1 |
| Pastoralist | 81 | 154 | 1.89(1.01, 3.56) | 0.68(0.22, 2.10) |
| Others | 21 | 35 | 2.16(0.98, 4.75) | 1.68(0.63, 4.47) |
| **Farming-land ownership** | | | | |
| Yes | 23 | 83 | 1 | 1 |
| No | 97 | 163 | 2.15(1.27, 3.63) | 1.77(0.98, 3.20) |
| **Maternal ANC follow-up** | | | | |
| Yes | 69 | 160 | 1 | 1 |
| No | 51 | 86 | 1.38(0.88, 2.15) | 0.85(0.48, 1.50) |
| **Gravidity** | | | | |
| Primi-gravida | 16 | 51 | 1 | 1 |
| 2–4 | 53 | 116 | 1.46(0.76, 2.79) | 0.48(0.05, 5.15) |
| ≥ 5 | 51 | 79 | 2.06(1.06, 3.99) | 0.39(0.03, 4.69) |
| **Birth interval** | | | | |
| First birth | 16 | 51 | 1.80(0.82, 3.94) | 1.02(0.10, 10.83) |
| < 36 month | 89 | 109 | 4.68(2.53, 8.67) | **4.80(2.41, 9.57)**$^*$ |
| ≥ 36 months | 15 | 86 | 1 | **1** |
| **Receiving nutrition counseling** | | | | |
| Yes | 66 | 162 | 1 | 1 |
| No | 54 | 84 | 1.58(1.01, 2.46) | 1.29(0.75, 2.20) |
| **Maternal minimum dietary diversity score** | | | | |
| Meet | 10 | 78 | 1 | **1** |
| Not meet | 110 | 168 | 5.11(2.53, 10.29) | **5.10(2.13, 12.24)**$^*$ |
| **Maternal minimum meal frequency** | | | | |
| Meet | 26 | 94 | 1 | 1 |
| Not meet | 94 | 152 | 2.24(1.35, 3.70) | 1.31(0.70, 2.43) |

$^*$significant at P-value<0.001

## Discussion

This study found that the level of undernutrition among lactating mothers is 32.8% (95% CI: 28, 38). This finding was similar to a study conducted in Genta Afeshum woreda of rural Tigray which reported 33.6% [30]. The insignificant difference might be the data collection period for the later study was conducted during Ethiopian Orthodox lent fasting period which might affect dietary feeding practice and nutritional status of the mother, while the study collection period for this study was not on the fasting period.

Compared to the findings of this study, a lower prevalence of undernutrition among lactating mothers was reported in Offa district, Wolyita Zone 15.8% [26], Moyale district, Borena Zone 17.7% [33], Ambo district, West Shewa Zone 21.5% [32], Wombera woreda 25.4% [23], Arba mich Zuria district 26.1% [39], and Raya-Alamata district (Lowland 17.5%, highland 24.6%) [29]. It is also higher than the studies from Nigeria [31] and Zambia [40], in which 4.7% and 3.4% of lactating mothers were undernourished, respectively. The possible reason might be due to the different socio-demographic nature of the area; this study is in the pastoralist community while the other studies were more agrarian.

On the contrary, the finding of this study is lower than the findings from studies done in the Eastern zone of Tigray 38% [34], and Jimma, 40.6% [22]. The possible discrepancy might due to the difference in agro ecologic and cultural variation of the study areas. The other explanation for the observed difference might be the participants included in a study done in Jimma were beneficiaries of the home-based food production and child-centered counseling project.

Maternal dietary diversity score and birth interval were showed a statistically significant association with undernutrition among lactating mothers. Based on this study, lactating mothers who didn't meet the MDDS were more than five times more likely to be underweight as compared to those who met the MDDS. This is consistent with studies conducted in Jimma [22], Borena [33], and Raya-Alamata districts [29]. This can be explained by dietary diversity is a proxy indicator of maternal nutrient adequacy, and this nutrient adequacy is essential for protein, lipid, and carbohydrate metabolism [41]. Maternal nutrient needs increase during lactation, and when these needs are not met, they may suffer from malnutrition. During lactation, the energy, protein, and other nutrients in breast milk come from a mother's diet or her body stores [42]. If lactating women do not get enough energy and nutrients they will be at risk of nutrient storage depletion and exacerbates undernutrition [43]. Monotonous diets full of starchy staples, low quantities of fruits and vegetables, and scant animal-source foods result in malnutrition [44, 45].

This study revealed that the probability of being undernutrition is more than four times among lactating mothers who had short birth intervals as compared to mothers who had a long duration of the birth interval. This finding is consistent with studies done in Nigeria [46, 47]. A birth interval of at least 36 months before couples deliver the next child is recommended for mothers and their children's health [48]. Lactation reduces the nutritional reserves of mothers, so longer inter-birth intervals allow for an increase in nutritional reserves [49].The short birth interval could not give time to the mother to restore her nutritional status, and expose the mother to nutritional stress, morbidity, and mortality [50, 51].

In this study, even though the socio-demographic variables have a significant association at bivariate analysis, they were not significantly associated in the final multivariable model. But some studies [6, 23, 24, 26, 32] showed that family size, maternal education, maternal occupation, farming land ownership, and paternal education and occupation showed a significant association with undernutrition among lactating mothers. Also, contrary to other studies [23] maternal health care characteristics such as teenage pregnancy, ANC follow-up, place of delivery, postnatal care follow-up, maternal meal frequency, nutrition-related education, and

maternal workloads had no statistically significant association with maternal undernutrition at multivariable logistic regression model. This difference is might be due to the difference in socio-demographic characteristics of the study areas. This study is conducted in a pastoralist community, while the other studies are mostly agrarian and with better access to health care services.

## Strengths of the study

Using primary data, and being a community-based study, which will estimate the true prevalence of the problem.

## Limitations of the study

Food consumption patterns might vary seasonally, and the results regarding dietary diversity score and meal frequency are limited to the specific season of the year in which the study was conducted. Those dietary assessments are also recorded using a 24-hour recall method; as a result, they might not reflect the usual intake of an individual. The other limitation of this study is failure to incorporate household food security and wealth index as factors associated with undernutrition which might have also introduced a residual confounding problem.

## Conclusions

In this study, nearly one-third of lactating mothers were undernourished, which is higher as compared to other similar studies, and considered as a significant public health problem. Undernutrition was significantly associated with maternal dietary diversity and birth interval. For the prevention of undernutrition among lactating mothers health education on proper and adequate maternal dietary feeding practices, and proper family planning utilization during lactation should be emphasized.

## Supporting information

**S1 File.**
(DOCX)

**S1 Data.**
(SAV)

## Acknowledgments

We acknowledge the Afar regional and woreda health office heads for their valuable cooperation during the data collection. We would like to extend our gratitude to the data collectors and lactating mothers who participated in this study. Finally, thanks to Samara University for ethical approval and facilitation of this study.

## Author Contributions

**Conceptualization:** Getahun Fentaw Mulaw.

**Data curation:** Getahun Fentaw Mulaw, Kusse Urmale Mare.

**Formal analysis:** Getahun Fentaw Mulaw, Kusse Urmale Mare.

**Investigation:** Getahun Fentaw Mulaw.

**Methodology:** Getahun Fentaw Mulaw, Kusse Urmale Mare.

**Resources:** Getahun Fentaw Mulaw.

**Supervision:** Getahun Fentaw Mulaw.

**Validation:** Etsay Woldu Anbesu.

**Visualization:** Getahun Fentaw Mulaw.

**Writing – original draft:** Getahun Fentaw Mulaw.

**Writing – review & editing:** Getahun Fentaw Mulaw, Kusse Urmale Mare, Etsay Woldu Anbesu.

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
