## [Decision Letter · Decision Letter 0]

12 Apr 2021

PONE-D-20-31073

Nearly One-Third of Lactating Mothers are Suffering from Under-nutrition in Pastoral Community, Afar Region, Ethiopia: Community-Based Cross-Sectional Study

PLOS ONE

Dear Dr. Mulaw,

Thank you for submitting your manuscript to PLOS ONE. After careful consideration, we feel that it has merit but does not fully meet PLOS ONE’s publication criteria as it currently stands. Therefore, we invite you to submit a revised version of the manuscript that addresses the points raised during the review process.

We look forward to receiving your revised manuscript.

Kind regards,

Charu C Garg, Ph.D.

Academic Editor

PLOS ONE

Additional Editor Comments:

Besides the comments from the reviewers please try to make the discussion more succinct, some of it is already mentioned in introduction  - it should be deleted or should be taken there eg the first para of the discussion. 

Please use PLOS One style to state the references in the text - use of square brackets

P. 3:” A semi-structured questionnaire was developed through a critical review of relevant literature.” Please mention the literature

Need to check all the Spellers and grammar – e.g p. 5 meat and not meet, multicollinearity etc. English editing is required. While the  nutrition outcome on two important variables  - dietary requirements and birth spacing are detailed, the discussion should also highlight how sociodemographic variables impact the outcomes. Could there be a problem of multi-collinearity, or they may be significant at higher p values. 

Pls comment on why BMI was preferred to other measures such as MUAC in some other studies. Was the association between MUAC and BMI studied.

Abbreviations should be explained whenever used first time in the text, instead of giving in the end.

Generally Ethical statement is a part of methodology.

Journal Requirements:

[Special thanks to Samara University for funding, and to study participants for providing needy information for this research.]

 [he funders had no role in study design, data collection and analysis, decision to publish, or preparation of the manuscript.]

5. Please include your tables as part of your main manuscript and remove the individual files. Please note that supplementary tables (should remain/ be uploaded) as separate "supporting information" files.

6. We suggest you thoroughly copyedit your manuscript for language usage, spelling, and grammar. If you do not know anyone who can help you do this, you may wish to consider employing a professional scientific editing service.  

The name of the colleague or the details of the professional service that edited your manuscriptA copy of your manuscript showing your changes by either highlighting them or using track changes (uploaded as a *supporting information* file)A clean copy of the edited manuscript (uploaded as the new *manuscript* file

7. Please include additional information regarding the survey or questionnaire used in the study and ensure that you have provided sufficient details that others could replicate the analyses. For instance, if you developed a questionnaire as part of this study and it is not under a copyright more restrictive than CC-BY, please include a copy, in both the original language and English, as Supporting Information. Moreover, please include more details on how the questionnaire was pre-tested, and whether it was validated.

8. In your Methods section, please provide additional information about the participant recruitment method and the demographic details of your participants. Please ensure you have provided sufficient details to replicate the analyses such as: a) the recruitment date range (month and year), b) a description of any inclusion/exclusion criteria that were applied to participant recruitment, c) a table of relevant demographic details, d) a statement as to whether your sample can be considered representative of a larger population, e) a description of how participants were recruited, and f) descriptions of where participants were recruited and where the research took place.

Reviewers' comments:

Reviewer's Responses to Questions

**Comments to the Author**

1. Is the manuscript technically sound, and do the data support the conclusions?

Reviewer #1: Yes

Reviewer #2: Yes

2. Has the statistical analysis been performed appropriately and rigorously? 

Reviewer #1: Yes

Reviewer #2: Yes

3. Have the authors made all data underlying the findings in their manuscript fully available?

Reviewer #1: No

Reviewer #2: Yes

4. Is the manuscript presented in an intelligible fashion and written in standard English?

Reviewer #1: Yes

Reviewer #2: No

5. Review Comments to the Author

Reviewer #1: Reviewer Comments

Journal: PLOS ONE

Title: Nearly One-Third of Lactating Mothers are Suffering from Under-nutrition in Pastoral Community, Afar Region, Ethiopia: Community-Based Cross-Sectional Study

ABSTRACT

Background part:

What means by the sentence ; ‘Lactating women from developing countries are nutritionally vulnerable groups because this period places a high nutritional demand on the mother and lead to nutritional stress’ i.e is to mean that lactating women from the developed country are not nutritionally vulnerable?

Methdology part:

The word methdology is not an appropriate word rather Methods

Study subjects are not described well i.e are all lactating mothers or lactating whose children less than 6 months or those who had children 6-24 months age included in your study is not stated vs types of anthropometric measurement used to determine nutritional status

What means by Study participants were recruited using a multi-stage sampling technique?

Which types of anthropometric measurements you used is not mentioned?

Types of instruments used for anthropometric measurement , sensitivity and specificity of instruments you used should be stated

General Comments: Your methods part needs some revision

Result part:

What means by, stunted (height < 145 cm) ? in stating of prevalence of your result part

BACKGROUND

Better if you begin your sentence with definition of your outcome variable

In your fourth paragraph of background section, what mean by the statement ‘significant proportion of lactating mothers are undernourished and maternal mortality ratio is still 412 per 100,000 live births’? how you relate this maternal mortality with under nutrition i.e Is that to mean that this maternal mortality is due to under nutrition???

General Comments: Your background section mainly focuses on statements of problem part but it lacks some important points about introduction (background) part.

Methods and materials

Study design, setting, and period part:

Nothing is mentioned about study period

Total population of your study subjects is not indicated

Eligibility criteria part:

You stated that respondents who were seriously ill and physically unable to fit for anthropometric measurements. What types of illness and physical deformity you found and excluded? It must be clearly stated.

How you confirm pregnancy to exclude from study?

Sample size determination and sampling procedure:

Why you used design effect 1.5?

You said that study participants were selected using a systematic sampling technique. I f you got the list of lactating mothers from health extension family folder, why you use systematic sampling technique?

Data collection procedure and quality control measures part:

you did not cite any literatures you used for the development of your questionnaire

What is the importance of using 10% of your sample size for pretesting?

Why you used nine food groups for calculation to determine WDDS??? Is it the right way using nine food groups to determine DDS for Women of Reproductive Age???

You said that duplicate anthropometric measurements were done in case of deviations from standard procedures. Which measurements (from this duplicate anthropometric measurements) you documented finally?

Data processing and analysis part:

The final model was tested using Hosmer and Lemeshow chi-square test p-value, and it was P=556, is that P=556???

Operational definitions part:

Maternal under-nutrition: Did you use only BMI to determine your outcome variables vs lactating mothers having less than 24 months age children???

Women dietary diversity score (WDDS): Number of food groups used ???

Minimum Meal frequency (MMF)- what is your reference/standard that to be said meal frequency as Meet: > 4 times and Not meet: < 4 times?

RESULT

Socio-demographic characteristics of study participants part:

Better if you include the response rate

More than half, 223(60.9%) of mothers were not attending formal education- better if you said three-fifth of….

Maternal nutritional status part:

Among the 366 respondents, 120(32.8%) of mothers were undernourished; with mild, moderate, and severe degrees of undernourishment occurred in 80(21.9%), 30(8.2%), and 10(2.7%) of respondents, respectively—Did you use only BMI to determine nutritional status i.e did you use BMI for lactating women whose children’s age less than 6 months ???

Discussion

In the fourth paragraph of your discussion, you said that ‘The possible reason for the difference might be the use of different measurement; they use MUAC while this study uses BMI’. What this sentence implies?

In the sixth paragraph of your discussion you said, A birth-interval of at least 36 months before couples deliver the next child is recommended for mothers and their children. Is 36 months or 24 months used as recommendation of birth interval in Ethiopian government? Did you use 36 months in your analysis?

In the last two paragraphs of your discussion part: please clearly explain the reason how/why this occurred (those variables which is significant in others study but not in your study) and vice versa.

Limitations of the study

Being a cross-sectional study- is it limitation? Did not know this (causal relationship) before conducting your study?

Fail to incorporate household food security assessment- Even if you stated as a limitation, this is a major issue that was not be missed especially in accordance with your study subjects and area.

Food Security-Major Issue

Reviewer #2: The researcher tried to address both the level of under nutrition and some associated factors among pastoralist population who are less advantages as compared the others settings. This is sound good!

However, the sample size determined was only for prevalence and we hesitate that this sample can determine the true predictors of undernutrition. Hence, Authors explanation is required to clear this hesitation.

The utilization grey literature like office unpublished report should be appropriately addressed based Vancouver system. The operational definition and measurement issues like Women dietary diversity score (WDDS) described on page 5 should be crystal clear for all readers based on scientific standard. This is also true for maternal nutritional status measurement described on the same page.

Finally, in discussion section some modification needed; particularly for the justifications narrated for discrepancy observed with regards to some findings.

6. PLOS authors have the option to publish the peer review history of their article (what does this mean?). If published, this will include your full peer review and any attached files.

Reviewer #1: No

Reviewer #2: **Yes: **Ashenafi Habtamu

---

## [Author Response · Author response to Decision Letter 0]

24 Apr 2021

Comments from the Editor: It was amazing to get such like professional comments from the editor; I accepted the comments and tried to amend them.

Comments from reviewer one: Thank you for your detail comments and suggestions. You have touched each part of the manuscript in detail, and accordingly I modified the whole document.

Comments from reviewer two: Thanks for constructive general comments, which indicate me to deal more on the manuscript

---

## [Decision Letter · Decision Letter 1]

21 Jun 2021

Nearly One-Third of Lactating Mothers are Suffering from Undernutrition in Pastoral Community, Afar Region, Ethiopia: Community-Based Cross-Sectional Study

PONE-D-20-31073R1

Dear Dr. Mulaw,

We’re pleased to inform you that your manuscript has been judged scientifically suitable for publication and will be formally accepted for publication once it meets all outstanding technical requirements.

Kind regards,

Charu C Garg, Ph.D.

Academic Editor

PLOS ONE

Additional Editor Comments (optional):

Thank you for taking into account the reviewers comments and addressing those. Most of the comments have been addressed, except a few which have been pointed out by one of the reviewer. I also have minor comments in terms of improving the grammar including the tenses. It would be better to consistently use the past tense. I have pointed some of these in the attached paper, though those are not comprehensive.

Reviewers' comments:

Reviewer's Responses to Questions

**Comments to the Author**

1. If the authors have adequately addressed your comments raised in a previous round of review and you feel that this manuscript is now acceptable for publication, you may indicate that here to bypass the “Comments to the Author” section, enter your conflict of interest statement in the “Confidential to Editor” section, and submit your "Accept" recommendation.

Reviewer #1: (No Response)

Reviewer #2: All comments have been addressed

2. Is the manuscript technically sound, and do the data support the conclusions?

Reviewer #1: Yes

Reviewer #2: Yes

3. Has the statistical analysis been performed appropriately and rigorously? 

Reviewer #1: Yes

Reviewer #2: Yes

4. Have the authors made all data underlying the findings in their manuscript fully available?

Reviewer #1: (No Response)

Reviewer #2: Yes

5. Is the manuscript presented in an intelligible fashion and written in standard English?

Reviewer #1: (No Response)

Reviewer #2: Yes

6. Review Comments to the Author

Reviewer #1: Reviewer Comments (After Revised )

Journal: PLOS ONE

Title: Nearly One-Third of Lactating Mothers are Suffering from Under-nutrition in Pastoral Community, Afar Region, Ethiopia: Community-Based Cross-Sectional Study

ABSTRACT

What is the importance of using two anthropometric measurements ( BMI and MUAC) to determine the same outcome (Under nutrition) in your research? i.e it is not clearly stated for whom you used BMI and for whom you used MUAC? Please clearly indicate this part.

Discrepancy in sampling technique i.e previously you said multi-stage sampling technique was used to recruit your study participants; but now, you said systematic sampling technique and you said that you got the list of your study subjects from health extension workers family folder. Do you think that you used the appropriate sampling technique in your case?

BACKGROUND PART

Better if you begin with definition of your outcome variable.

Methods and materials

Study design, setting, and period part

Better if you included the total number of your study subjects (lactating women) from the total populations in the district

Eligibility criteria part

In your exclusion criteria, still how you confirmed pregnancy is not clear???

You stated that ‘Mothers were considered to be excluded if they were seriously ill and physically unable to fit for anthropometric measurements, but no lactating mother was found with such cases’. So if you did not get such cases, no need of stating this in your exclusion criteria.

Sample size determination part

Previously, you said that you used design effect of 1.5 and I said why you used 1.5? But now you removed this; removing questions from your paper does not mean you correct it. I want to know how you did this???

Sampling technique you used to select your participants is not correct

Operational Definitions

Your operational definition of Under nutrition needs a reference and better if you writing in a more operationalized way

MDDS- food groups you used to classify this is not the right ( It was better if you used the current standard). So it may better if you include in your limitation part. The other issue here is that your operationalized here ( MDDS) is not matched as you present it in your result part

Discussion

Better if you begin with aims/importance of your study ( say something as introduction )

Reason of explanation for the existing differences are not sufficiently addressed

(Especially, 1st , 2nd and last paragraphs)

Reviewer #2: The authors tried to address the comments given in the previous version. Hence, now it look good in its current version

7. PLOS authors have the option to publish the peer review history of their article (what does this mean?). If published, this will include your full peer review and any attached files.

Reviewer #1: **Yes: **Hunegnaw Almaw Derseh

Reviewer #2: No

---

## [Editor Report · Acceptance letter]

1 Jul 2021

PONE-D-20-31073R1 

Nearly One-Third of Lactating Mothers are Suffering from Undernutrition in Pastoral Community, Afar Region, Ethiopia: Community-Based Cross-Sectional Study 

Dear Dr. Mulaw:

I'm pleased to inform you that your manuscript has been deemed suitable for publication in PLOS ONE. Congratulations! Your manuscript is now with our production department. 

Kind regards, 

on behalf of

Dr. Charu C Garg 

Academic Editor

PLOS ONE